# In Search of Robust Measures of Generalization

**Gintare Karolina Dziugaite**[1*]**, Alexandre Drouin**[1*]**, Brady Neal**[1,2,3]**, Nitarshan Rajkumar**[2,3]**,**

**Ethan Caballero**[2,3]**, Linbo Wang**[4]**, Ioannis Mitliagkas**[2,3]**, Daniel M. Roy**[4,5]

[1]Element AI, [2]Mila, [3]Université de Montréal, [4]University of Toronto, [5]Vector Institute

## Abstract

One of the principal scientific challenges in deep learning is explaining generalization, i.e., why the particular way the community now trains networks to achieve small training error also leads to small error on held-out data from the same population. It is widely appreciated that some worst-case theories—such as those based on the VC dimension of the class of predictors induced by modern neural network architectures—are unable to explain empirical performance. A large volume of work aims to close this gap, primarily by developing bounds on generalization error, optimization error, and excess risk. When evaluated empirically, however, most of these bounds are numerically vacuous. Focusing on generalization bounds, this work addresses the question of how to evaluate such bounds empirically. Jiang et al. [9] recently described a large-scale empirical study aimed at uncovering potential causal relationships between bounds/measures and generalization. Building on their study, we highlight where their proposed methods can obscure failures and successes of generalization measures in explaining generalization. We argue that generalization measures should instead be evaluated within the framework of distributional robustness.

## 1 Introduction

Despite tremendous attention, a satisfying theory of generalization in deep learning remains elusive. In light of so many claims about explaining generalization in deep learning, this statement is somewhat controversial. It also raises an important question:

*What does it mean to explain generalization in deep learning?*

In this work, we propose empirical methodology to aid in the search of a precise mathematical theory, allowing us to leverage large-scale empirical studies of generalization, like those in recent work [8, 9]. Unlike earlier work, however, our proposal rests on the foundation of *robust prediction*, in order to catch out, rather than average out, failures.

The dominant approach to studying generalization is the frequentist framework of statistical learning theory. We focus our attention on the simplest setting within supervised classification, where the training data, $S$, are modeled as a sequence of $n$ random variables, drawn i.i.d. from a distribution $\mathcal{D}$ on labeled examples $(x, y)$. In supervised classification, learning algorithms choose a classifier $h_S$, based on the training data $S$. Ignoring important considerations such as fairness, robustness, etc., the key property of a classifier $h$ is its probability of error, or (classification) risk,

$$R_{\mathcal{D}}(h) = \mathbb{P}_{(x,y)\sim\mathcal{D}}[h(x) \neq y]. \tag{1}$$

One of the key questions is why deep learning often produces classifiers with human-level risk in domains that stymied researchers for decades. In this work, we take an empirical perspective and

judge theories of generalizations by the predictions they provide when tested. In the other direction, any systematic rule for predicting generalization—whether learned or invented—can be thought of as a theory that can be tested.

We consider families of environments, defined by data distributions, architectural choices, train set sizes, learning algorithms and their tuning parameters, etc. Given a particular family of environments, a strong theory achieves a desired level of precision for its predictions, while depending as little as possible on the particular details of the environments. At one extreme, explanations based on the VC dimension of the zero–one loss class of neural networks would pin the success of deep learning on empirical risk minimization. In practice, these explanations are poor, not just because the ensuing bounds are numerically vacuous for the size of networks and datasets used in practice, but because they fail to correctly predict the effect of changes to network width, depth, etc.

At the other extreme, average risk on held-out data (i.e., a test-set bound) provides a sharp estimate of risk, yet the computation to produce this estimate is inextricably tied to every detail of the learned weights and data distribution. Viewing predictors as theories, the test-set bound is essentially silent. Any satisfactory theory of generalization in deep learning must therefore lie between these two extremes. We must necessarily exploit properties of the data distribution and/or learning algorithm, but we must also be willing to trade precision to decrease our dependence on irrelevant details.

What dependence on the data distribution and learning algorithm is necessary to explain deep learning? Even taking the data distribution into consideration, the fact that stochastic gradient descent (SGD) and its cousins often perform empirical risk minimization cannot explain generalization [29]. There is, however, a picture emerging of overparametrization and SGD conspiring to perform capacity control. In some theoretical scenarios, this control can be expressed in terms of norms. At the same time, great strides have been made towards identifying notions of capacity that can be shown to formally control generalization error, $R_{\mathcal{D}}(h) - \hat{R}_S(h)$, uniformly over $h$ belonging to specially defined classes. (Here $\hat{R}_S(h)$ denotes the empirical risk, as estimated by the training data.)

Despite this progress, there is still a wide gulf between performance as measured empirically via held-out data and performance as predicted by existing theory. No expert would be surprised to discover that a published bound yields predictions for risk that are numerically vacuous when evaluated empirically. A standard retort is that the constants in most bounds are suboptimal or that the purpose of bounds is to identify qualitative phenomena or inspire the development of new algorithms. Even ignoring the issue of numerically vacuous bounds, many bounds demonstrably fail to account for the right dependencies. As a case in point, recent empirical work [17] identifies state-of-the-art bounds on generalization error that grow with training set size, while the generalization error actually shrinks. Indeed, many bounds are distribution- or data-dependent and so the question of whether they explain generalization in practice is an empirical question.

## 1.1 Large-scale empirical studies

Recent work proposes large-scale empirical investigations to study generalization [9]. (See also [8].) While it is becoming more common for theoretical work to present empirical evaluations, among recent empirical studies [2, 3, 5, 11, 12, 19, etc.], most are limited. One motivation for large-scale empirical studies is to leverage massive computing resources in the pursuit of a scientific challenge that has largely been approached mathematically. Another motivation is to go beyond simply measuring correlation towards measuring aspects of causation. (Several authors of this work—Dziugaite, Neal, and Roy—have each advocated for this publicly.)

Given how influential these proposals by Jiang et al. [9] may be, we believe they deserve critical attention. (Indeed, recent preprints have already started to integrate their methodology.) Jiang et al. [9] propose to use Kendall correlation coefficients and independence tests to evaluate a suite of so-called *generalization measures*. Many of these generalization measures are frequentist bounds, though with missing constants or lower-order terms. Others are only loosely inspired by bounds.

One of central proposals made by Jiang et al. is to *average* evaluation metrics (i.e., Kendall-$\tau$) over a range of experimental settings, which arise from targeted changes to hyperparameters.[2] In contrast, we argue that averaging across experimental settings is *not* an appropriate summarization of the strength of a generalization measure as a theory of generalization. In particular, a satisfying

theory should admit a generalization measure that offers reasonable predictions of generalization across the entire range of experimental settings under study. A theory—realized by a generalization measure—is as strong as its weakest part: *a satisfying theory cannot simply predict well on average*.

The study of prediction across a range of environments is the subject of *distributional robustness* [1, 4]. An extreme form of robustness is obtained when one seeks to predict well in all environments that may arise from all possible interventions to an experimental setting. This extreme form of robustness can be linked to a weak form of causality [4, 16].

Crucially, we do not aim for robustness over all possible environments. To achieve some level of generality, useful theories must necessarily have limited scope. As we demonstrate in Section 2, frequentist generalization bounds can exploit noncausal correlations that can be seen to stand in for unknown properties of the data distribution because of properties of the learning algorithm. Such bounds have an important role to play in our search for a theory of generalization in deep learning, but we cannot expect them to explain generalization under interventions that upset these noncausal correlations. Bounds that depend on properties of the data distribution have an important role to play, though one hindered by the statistical barriers of unknown distributions, accessible only through a limited pool of data. More general theories (that minimize this dependence) can pinpoint key data properties.

**Contributions.** Theories of generalization yield predictions: How should we evaluate these predictions empirically? In this work, we adopt the proposal of [9] to exploit large-scale empirical studies, but critique the use of averaging in the evaluation of predictions made by these theories. Based on the specific scientific goals of understanding generalization, we propose that the framework of distributional robustness is more appropriate, and suggest how to use it to evaluate generalization measures.

Besides theoretical contributions, we make empirical contributions: We collect data from thousands of experiments on CIFAR-10 and SVHN, with various values for width, depth, learning rate, and training set size. We adopt the ranking task and sign-error loss introduced by [9], but use the collected data to perform a *robust* ranking evaluation, across the 24 candidate generalization measures on over 1,600,000 pairs of experiments.

We find that *no existing complexity measure has better robust sign-error than a coin flip*. Even though some measures perform well on average, every single measure suffers from 100% failure in predicting the sign change in generalization error under some intervention. This observation is not the end of the evaluation, but the beginning.

To better understand the measures, we evaluate them in families of environments defined by interventions to a single hyperparameter. We find: (i) most, though not all, measures are good at robustly predicting changes due to training set size; (ii) robustly predicting changes due to width and depth is hard for all measures, though some PAC Bayes-based measures are robust across a large fraction of the environments tested; (iii) norm-based measures outperform other measures at learning rate interventions.

By focusing on robust evaluation, we force ourselves to dig into the data to uncover the cause of failures—failures which might otherwise go undiscovered by looking at average performance. As such, robustness provides better guidance to the scientific challenge of explaining generalization. The rest of this paper is organized as follows. In Section 2, we present a concrete example of a frequentist analysis of a learning algorithm, which reiterates some of the high-level points above. We then introduce distributional robustness in Section 3 and describe how the framework can be used to analyze large-scale empirical studies of generalization measures in Section 4. We detail our experimental setting in Section 5 and summarize our experimental findings in Section 6 before ending with a discussion.

## 2 A Motivating Example: SVMs, Norm-based Capacity Control, and the Role of Causality

In this section, we study support vector machines (SVMs) to demonstrate some of the challenges in understanding and explaining generalization error. This section owes much to [27]. The intuition extracted from this simple model motivates our methodological choices for the rest of this paper. In

particular, we see that frequentist generalization bounds derived under one set of conditions may rely on quantities that do not have a direct causal relationship with the generalization error under other conditions. This highlights that frequentist bounds can be expected to have limits to the predictive powers under intervention, but also that asking for causal measures of generalization may rule out measures that nonetheless work well in a range of scenarios.

Consider linear prediction, based on an embedding of inputs into $\mathbb{R}^p$. As usual, we index the space of linear predictors by nonzero vectors $\boldsymbol{w} \in \mathcal{H} = \mathbb{R}^p$, where the decision boundary associated to $\boldsymbol{w}$ is the tangent hyperplane $\{\boldsymbol{x} \in \mathbb{R}^p : \langle \boldsymbol{w}, \boldsymbol{x} \rangle = 0\}$, passing through the origin. Assuming labels take values in $\{\pm 1\}$, the zero–one classification loss of the predictor $\boldsymbol{w}$ on a labeled input $(\boldsymbol{x}, y)$ is $\ell(\boldsymbol{w}, (\boldsymbol{x}, y)) = \frac{1}{2}(1 + y \, \mathrm{sgn}(\langle \boldsymbol{x}, \boldsymbol{w} \rangle))$. Note that the loss is invariant to the magnitude of the vector $\boldsymbol{w}$, and so the set of hyperplanes can be put into correspondence with the unit vectors $\overline{\boldsymbol{w}} := \boldsymbol{w}/\|\boldsymbol{w}\|$. We focus on the realizable setting, i.e., data are assumed to be labeled according to some hyperplane. In this case, every finite data set admits a positive cone of empirical risk minimizers.

Let $S = \{(\boldsymbol{x}_i, y_i)\}_{i \in [n]}$ be $n$ i.i.d. labeled data in $\mathbb{R}^p \times \{\pm 1\}$, and let $\boldsymbol{w}_S$ be chosen according to the SVM rule: $\min_{\boldsymbol{w}} \|\boldsymbol{w}\|^2$, subject to the constraint that $y_i \langle \boldsymbol{x}_i, \boldsymbol{w} \rangle \geq 1$ for all $i \in [n]$. The constraint demands that, for each data point, the functional margin, $y_i \langle \boldsymbol{x}_i, \boldsymbol{w}_S \rangle$, be greater than one. Thus the hyperplane $\overline{\boldsymbol{w}_S}$ indeed separates the data and achieves zero empirical risk. However, among the vectors that satisfy the margin constraint, $\boldsymbol{w}_S$ has the smallest L2 norm. Geometrically, the hyperplane $\overline{\boldsymbol{w}_S}$ is that with the largest *geometric* margin, $\min_i y_i \langle \boldsymbol{x}_i, \overline{\boldsymbol{w}_S} \rangle$.

Why does the SVM classifier generalize? The best explanation may depend on the situation. The VC dimension of the space of $p$-dimensional linear predictors is $p$, and so, with high probability over the sample $S$, uniformly over all separating hyperplanes $\boldsymbol{w}$, the difference between the empirical risk and risk is $\tilde{O}(p/n)$. If $n \gg p$, then this reason alone suffices to explain strong performance. The details of the SVM rule are irrelevant beyond it returning an empirical risk minimizer.

Suppose that we consider a family of embeddings of growing dimensionality and find that the SVM rule generalizes equally well across this family. The VC theory cannot explain this. A theory based on the maximum-margin property of the SVM rule may. To that end, assume there exists a hyperplane $\boldsymbol{w}_*$ such that $y \langle \boldsymbol{w}_*, \boldsymbol{x} \rangle \geq 1$ with probability one over the pair $(\boldsymbol{x}, y)$. To fix a scale, assume $\|\boldsymbol{x}\| \leq \rho$ with probability one. By exploiting strong convexity, and the fact that $\|\boldsymbol{w}_S\| \leq \|\boldsymbol{w}_*\|$, one can show that the risk of $\boldsymbol{w}_S$ is bounded by $\tilde{O}(\rho\|\boldsymbol{w}_*\|/n)$. Note that this bound has *no explicit dependence* on the dimension $p$. Instead, it depends on the quantity $\rho\|\boldsymbol{w}_*\|$, whose reciprocal has a geometric interpretation: the distance between the separating hyperplane $\overline{\boldsymbol{w}_*}$ and the nearest data point, normalized by the radius of the data. Therefore, this analysis trades dependence on dimension for dependence on the data distribution's density near the decision boundary. When $\rho\|\boldsymbol{w}_*\| \ll n$, SVM's inductive bias is sufficient to explain generalization, even if $p \gg n$.

In fact, we can always build a bound based on the norm of the learned weights: with high probability, for *every* ERM $\boldsymbol{w}_S$, the risk is bounded by $\tilde{O}(\rho\|\boldsymbol{w}_S\|/n)$. One might prefer such a bound since $\|\boldsymbol{w}_*\|$ is often presumed unknown. Even if this bound matches risk empirically, it has a strange property: The bound depends on the norm $\|\boldsymbol{w}_S\|$ even though risk is *independent of norm*. Thus, we cannot expect the bound to remain valid if we intervene on the norm after training, e.g., to test for a causal relationship between norms and risk. Norms are the effect of the data and SVM interacting.

This example highlights that there may be multiple overlapping explanations depending on the range of environments in which one wants to understand generalization. We cannot, however, expect a theory to be robust to arbitrary interventions. Identifying a theory with limitations may lead us to more general ones, once we understand those limitations. All of this motivates a careful design of experimental methodology, in order to navigate these tradeoffs. In particular, we demand that a theory is *robustly predictive* of generalization over a carefully designed family of *environments*.

## 3 Preliminaries on Robust Prediction

In this section, we introduce the framework of robust prediction, borrowing heavily from Bühlmann [4], Peters, Bühlmann, and Meinshausen [23], and Rothenhäusler et al. [25]. In the next section, we cast the problem of studying generalization into this framework.

Consider samples collected in a family $\mathcal{F}$ of different *environments*. In particular, let $(\Omega, \mathcal{A})$ denote a common (measurable) sample space and, in each of these environments $e \in \mathcal{F}$, assume the data

we collect are drawn i.i.d. from a distribution $P^e$ on $\Omega$. We will think of environments as representing different experimental settings, interventions to these experiments, sub-populations, etc. For example, each sample might be a covariate vector and binary response, i.e., $\Omega = \mathbb{R}^p \times \{0, 1\}$. A well-studied setting is where the distributions $P^e$ all agree on the conditional mean of the response given the covariates (i.e., the regression function), but disagree on the distribution of the covariates.

Prediction is formalized by a *loss function*. In particular, a loss function for a set $\Phi$ of predictors is a map $\ell : \Phi \times \Omega \to \mathbb{R}$. The *error* or *risk* (of a predictor $\phi \in \Phi$ in an environment $e \in \mathcal{F}$) is then the expected loss, $\mathbb{E}_{\omega \sim P^e}[\ell(\phi, \omega)]$. If we focus on one environment $e \in \mathcal{F}$, it is natural to seek a predictor $\phi \in \Phi$ with small risk *for that individual environment*. However, if we care about an entire family $\mathcal{F}$ of environments, we may seek a predictor that works well simultaneously across $\mathcal{F}$. In the setting of *distribution robustness*, the performance of a predictor relative to a family $\mathcal{F}$ of environments is measured by the *robust error (or risk)*

$$\sup_{e \in \mathcal{F}} \mathbb{E}_{\omega \sim P^e}[\ell(\phi, \omega)]. \qquad (2)$$

The goal of robust prediction is to identify a predictor with small robust risk.

**Connection to causality.**    If taken to an extreme, then robust prediction is closely related to learning causality. Specifically, suppose that $(X, Y)$ is induced by a common causal model $Y := f(X)$. If $\mathcal{F}$ represents all possible interventions on subsets of $X$, then the causal predictor $f(X)$ also minimizes the robust risk. See [4, 24] for more details.

# 4    Studying Generalization via Distributional Robustness

We are interested in understanding the effects of changes to a complex machine learning experiment, with a focus on effects on generalization. In this section, we cast this problem into the framework of distributional robustness. In order to study generalization, we view theories of generalization as yielding predictors for generalization under a range of experimental settings. We use the term *generalization measure* to refer to such predictors.

## 4.1    Experimental Records and Settings

In the notation of Section 3, points $\omega \in \Omega$ represent possible samples. In our setting, each sample represents a complete record of a machine learning experiment. An environment $e$ specifies a distribution $P^e$ on the space $\Omega$ of complete records.

In the setting of supervised deep learning, a complete record of an experiment would specify hyperparameters, random seeds, optimizers, training (and held out) data, etc. Ignoring concerns of practicality, we assume the complete record also registers every conceivable derived quantity, not only including the learned weights, but also the weights along the entire trajectory, training errors, gradients, etc. Formally, we represent these quantities as random variables defined on the probability spaces $(\Omega, \mathcal{A}, P^e)$, $e \in \mathcal{F}$. Among these random variables, there is the empirical risk $\hat{R}$ and risk $R$ of the learned classifier, and their difference, $G$, the generalization error/gap.

Each distribution $P^e$ encodes the relationships between the random variables. Some of these relationships are common to all the environments. E.g., the generalization error $G$ always satisfies $G = R - \hat{R}$, and the empirical risk $\hat{R}$ is always the fraction of incorrectly labeled examples in the training data. Some relationships may change across environments. E.g., in a family $\mathcal{F}$ designed to study SGD, changes to, e.g., the learning rate, affect the distribution of the trajectory of the weights.

In machine learning, environments arise naturally from learning algorithms applied to benchmarks under standard hyperparameter settings. In order to evaluate theories that explain the effect of, e.g., hyperparameter changes, we also consider environments arising from perturbations/interventions to standard settings. E.g., we may modify the hyperparameters or data, or intervene on the trajectory of weights in some way. Every perturbation $e$ is captured by a different distribution $P^e$.

With respect to a family of environments $\mathcal{F}$, a generalization measure is preferred to another if it has smaller robust error (2). In Sections 5 and 6, we restrict our attention to $\mathcal{F}$ induced by varying hyperparameters, data distributions, training datasets, and dataset sizes. In this work, we do not intervene on the dynamics of SGD. However, intervening on the trajectory induced by SGD might be an interesting future direction that could allow one to tease apart the role of implicit regularization.

## 4.2 Prediction tasks

The predictions associated with a theory of generalization are formalized in terms of a map $C : \Omega \rightarrow \mathbb{R}$, which we call a *generalization measure*. We will study ad hoc generalization measures as well as ones derived from frequentist bounds. In both cases, we are interested in the ability of these measures to predict changes in the generalization.

One important aspect of a generalization measure is the set of (random) variables (i.e., covariates) it depends on. Indeed, there is an important difference between the task of predicting generalization using only the architecture and number of data and using also, e.g., the learned weights. Formally, let $V_1, \ldots, V_k$ be a finite collection of random variables. A generalization measure $C$ is $\sigma(V_1, \ldots, V_k)$-*measurable* if there exists a map $g$ such that $C(\omega) = g(V_1(\omega), \ldots, V_k(\omega))$ for all $\omega \in \Omega$. We may prefer one generalization measure to another on the basis of the covariates it uses. As a simple example, if a generalization measure offers comparable precision to another measure, but is measurable with respect to a strict subset of variables, then this increased generality may be preferred.

**Goals of the prediction.** We are broadly interested in two types of prediction tasks, distinguished by whether we train one or two networks.

In *coupled-network* experiments, we train two networks, such that they share all hyperparameters except one. We are interested in trying to predict which network has smaller generalization error.

Some of the generalization measures we consider are based on generalization bounds from the literature. Given that generalization bounds are often numerically vacuous, it would not be informative to evaluate their predictions directly at this stage. It is, however, reasonable to evaluate whether they capture the right dependencies. Indeed, one desirable property of evaluating generalization measures by the rankings they induce in coupled-network experiments is that the rankings are invariant to monotonically increasing transformations of the measure.

In *single-network* experiments, we try to predict the numerical value of the generalization error for that network based on a linear or affine function of a generalization measure. Generalization measures that perform well in such a task would serve as accurate predictors of generalization, and could be used for, e.g., model selection. However, such measures would not necessarily serve to be useful in generalization bounds. We describe the experimental details and results of *single-network* experiments in Appendix B due to space limitations.

## 5 Experimental methodology

In coupled-network experiments, we evaluate the *ranking* that the generalization measure induces on training networks. The approach we describe here is a robust analogue of the Kendall-$\tau$-based approach advocated by Jiang et al. [9].[3] This change is deceptively minor. We highlight the very different conclusions drawn using our methodology in Section 6.

**Evaluation criterion.** In more detail, recall that a coupled-network environment $e$ determines a distribution $P^e$ on pairs $(\omega, \omega')$ of *variable assignments*, each representing a full record of an experiment. We evaluate a generalization measure, $C$, and the realized generalization error, $G$, on both assignments, $\omega$ and $\omega'$. We use the ranking of $C$ values to predict the ranking of $G$ values. Then, the *sign-error of a generalization measure $C$ for this task*[4] is given by

$$\text{SE}(P^e, C) = \tfrac{1}{2} \mathbb{E}_{(\omega, \omega') \sim P^e} \left[ 1 - \text{sgn}\left(G(\omega') - G(\omega)\right) \cdot \text{sgn}\left(C(\omega') - C(\omega)\right) \right]. \tag{3}$$

Given a family $\mathcal{F}$ of coupled-network environments, the *robust sign-error of a generalization measure $C$* is $\sup_{e \in \mathcal{F}} \text{SE}(P^e, C)$. The $\Psi$ summary proposed by Jiang et al. [9] is analogous to the average sign-error, $|\mathcal{F}|^{-1} \sum_{e \in \mathcal{F}} \text{SE}(P^e, C)$.[5]

In our experiments, we use a modification of the loss in Eq. (3) in order to account for Monte Carlo variance in empirical averages. We use a weighted empirical average, where the weight for a sample

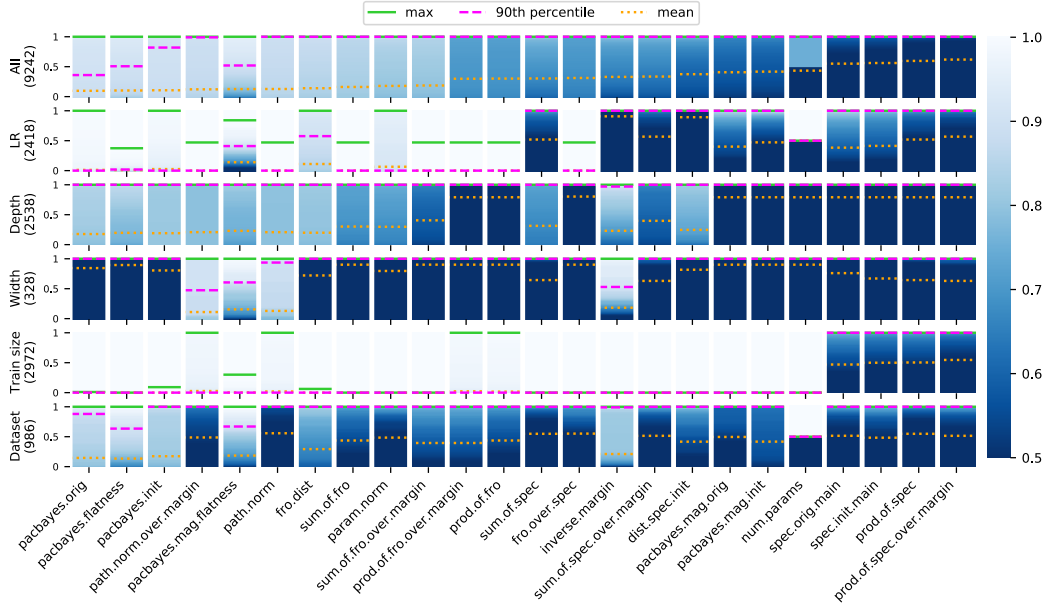

Figure 1: Cumulative distribution of the sign-error across subsets of environments for each generalization measure. The measures are ordered based on the mean across 'All' environments. A completely *white* bar indicates that the measure is perfectly robust, whereas a *dark blue* bar indicates that it completely fails to be robust.

$(\omega, \omega')$ is calculated based on the difference in generalization error $|G(\omega) - G(\omega')|$. We discard samples for which the difference in generalization error is below the Monte Carlo noise level. In effect, we control the precision to which we want our generalization measure to predict changes: when the difference is insignificant, we do not predict the sign. See Appendix A for the details on how we use the Monte Carlo variance to choose what environments are being considered. Other details of data collection are described in Appendix C.

**Environments.** In our experiments, variable assignments $(\omega)$ are pairs $(H, \sigma)$ of hyperparameter settings and random seeds, respectively. The hyperparameters are: learning rate, neural network width and depth; dataset (CIFAR-10 or SVHN), and training set size. (See Appendix C for ranges.)

Each environment $e$ is a pair $(H, H')$ of hyperparameter settings that differ in the setting of *one* hyperparameter (e.g., depth changes from $2 \rightarrow 3$ between $H$ and $H'$ and the remaining hyperparameters are identical). The distribution $P^e$ for a pair $e = (H, H')$ is the distribution of $(\omega, \omega') = ((H, \sigma), (H', \sigma'))$, where the random seeds $\sigma, \sigma'$ are chosen uniformly at random. That is, the expectation in Eq. (3) is taken only over a random seed.

## 6 Empirical Findings

In Fig. 1, we present a visualization of 1,600,000 ranking evaluations on 24 generalization measures derived from those used in [9]. A full description of these measures can be found in Appendix C.6. Motivated by the discussion in the introduction, we seek strong predictive theories: generalization measures that increase monotonically with generalization error and for which this association holds across a range of environments. Such a measure would achieve zero robust sign-error (Eq. (3)).

As described in Section 5, each environment contains a pair of experiments that share all hyperparameters but one (learning rate, depth, width, train set size, dataset). In each environment, we calculate the weighted empirical average version of the sign-error over 100 samples from $P^e$ (10 networks runs with different seeds per $\omega$. Note that we discard environments where too many samples have differences in generalization error below the Monte Carlo noise level (see Appendix A.2

for details). This is in contrast with the protocol proposed by Jiang et al. [9] where such noise is not filtered and can significantly undermine the estimation of sign-error (see Appendix A.3).

In the remainder of this section we interpret the results of Fig. 1, highlight some significant shortcomings of the generalization measures, and point out cases where these shortcomings would have been obscured by non-robust, average-based summary statistics like those used by Jiang et al. [9].

**How to read Fig. 1.** This figure presents the empirical cumulative distribution function (CDF) of the sign-error across all environments and generalization measures. **Every row** shows the CDF over a subset of environments (e.g., those where only depth is varied). The 'All' row shows the same but over all environments. The number of environments in each subset is given on the left of each row. **Each bar** in the figure is the empirical CDF of all sign-errors in the set of environments. A bar's y-axis corresponds to the range of possible sign-errors and the internal coloring depicts the distribution (starting at the median value for improved readability). We annotate the bars with the max (i.e., robust sign-error; **green**), the 90th percentile (**magenta**), and the mean (**orange**). The latter statistics do not measure robustness over all environments. However, a low 90th percentile value means the measure would have had low empirical robust sign-error restricting to some 90% of the environments tested. If the max is at 1.0, then there exists at least one environment where the measure fails to predict the sign of the change in generalization on all random seeds. If the max is below 0.5, then the measure is more likely than not to predict the correct sign on *all environments* in the set. *Identifying subfamilies in which a measure is robust is one of our primary objectives.*

**No measure is robust.** As illustrated in the 'All' row, for every one of the 24 measures, there is at least one environment in which the measure *always incorrectly predicts* the direction of change in generalization. Nonetheless, some measures have low robust error over large fractions of environments, as reflected by the 90th percentiles of the sign-error distributions. Notice how the average-based summaries proposed by Jiang et al. [9] do not reflect robustness, which implies their inability to detect the causal associations that they seek. Given these poor results, we must dig deeper to understand the merits and shortcomings of these generalization measures. Therefore, we study their performance in natural subfamilies $\mathcal{E} \subseteq \mathcal{F}$ of environments. Our analyses of the 'Train Size', 'Depth', and 'Width' rows below are examples of this. While no measure is robust across the CIFAR-10 and SVHN datasets considered here, we find measures that are quite robust over a 90% fraction of environments for SVHN only (see Appendix D.1).

**Robustness to train set size changes is not a given.** In the 'Train size' row, most measures correctly predict the effect of changing the train set size. (In general, generalization error decreases with train set size.) It may seem a foregone conclusion that a bound of the form $\tilde{O}(\sqrt{c/n})$ would behave properly, but, for most of these measures, the complexity term $c$ is a random variable that can grow with more training data. In fact, while many measures do achieve a low robust sign-error, some measures fail to be robust. In particular, some bounds based on Frobenius norms (e.g., `prod.of.fro`; Appendix C.6.4 and [20]) increased with train set size in some cases. Such corner cases arose mostly for shallow models (e.g., depth 2) with limited width (e.g., width 8) and were automatically identified by our proposed method. Note that the same finding was recently uncovered in a bespoke analysis [17], and we may have missed this looking only at average sign-errors, which are usually low.

**Robustness to depth.** In the 'Depth' row, we depict robust sign-error for interventions to the depth. Again, robust sign-error is maxed out for every measure. Digging deeper, these failures are not isolated: many measures actually fail in most environments. However, there are exceptions: a few measures based on PAC-Bayes analyses show better performance in some environments. In Fig. 2, we dig into the performance of `pacbayes.mag.flatness` (Appendix C.6.6) by looking at the subset of environments where it performs well (e.g., varying depth $3 \rightarrow 4$), fails but shows signs of robustness (e.g., $3 \rightarrow 6$), completely fails (e.g., $4 \rightarrow 5$), and those were a conclusion cannot be reached (e.g., $5 \rightarrow 6$). Looking into the data, we found that almost all environments where the measure fails are from the CIFAR-10 dataset, where the smaller networks we test suffer from significant overfitting. This illustrates how our proposed methodology can be used to zero-in on the limited scope where a measure is robust.

**Robustness to width is surprisingly hard.** In the 'Width' row, all measures have robust sign-error close to 1. Looking into the data, we discover that generalization error changes very little in response to interventions on width because the networks are all very overparametrized. In fact, of the 4,000 available width environments, only 328 remain after accounting for Monte Carlo noise.

**Comparison to Jiang et al. [9]** Our contribution is primarily a methodological refinement to the proposals in [9]. We describe how to discover failures of generalization measures in specific environments by looking at worst-case rather than average performance. We note that there are several reasons that even our *average-case* results are not directly comparable with those in [9]. First, their analysis considers the CIFAR-10 and SVHN datasets in isolation, whereas we combine models trained on both datasets. Second, they do not account for Monte Carlo noise, which we found to significantly alter the distribution of sign-errors (see Appendix A.3). This is important since we found that many environments had to be discarded due to high noise (e.g., only 8.2% of the width environments remain after filtering out noise in our analysis). Third, the hyperparameters and ranges that they consider are different from ours and, consequently, both studies look at different populations of models. For example, the majority of models in [9] use dropout, whereas our models do not. Such differences can alter how generalization measures and gaps vary in response to interventions on some hyperparameters and lead to diverging conclusions. For instance, in our results, no measure has an average-case performance much better than a coin-flip in the 'Depth' environments for CIFAR10, while Jiang et al. [9] find measures that perform well in this context. Nevertheless, there are some general findings that persist across both

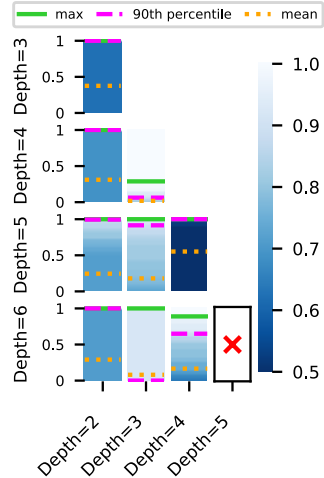

Figure 2: CDFs of sign-errors separated by environments where depth varies between two values for `pacbayes.mag.flatness`. The red X indicates that no environments remained after accounting for Monte Carlo noise.

studies; for instance, we see the good average-case performance of the path-norm (Appendix C.6.5) and PAC-Bayes-flatness-based (Appendix C.6.6) measures in contrast to the poor performance of spectral measures (e.g., `prod.of.spec`; Appendix C.6.3). We also find more specific similarities, such as the poor average-case performance of most measures in 'Width' environments for CIFAR-10 (Appendix D.2), in contrast to the better performance of `path.norm` (Appendix C.6.5), `path.norm.over.margin` (Appendix C.6.5), and `pacbayes.mag.flatness` (Appendix C.6.6).

## 7 Discussion

The quest to understand and explain generalization is one of the key scientific challenges in deep learning. Our work builds on recommendations in [9] to use large-scale empirical studies to evaluate generalization bounds. At the same time, we critique some aspects of these recommendations. We feel that the proposed methodology in [9] based on taking averages of sign-errors (or independence tests, which we have not pursued) can obscure failures. Indeed, for a long time, empirical work on generalization has not been systematic, and as a result, claims of progress outpace actual progress.

Based on an understanding of the desired properties of a theory of generalization, we propose methodology that rests on the foundation of distributional robustness. Families of environments define the range of phenomena that we would like the theory to explain. A theory is then only as strong as its worst performance in this family. In our empirical study, we demonstrated how a family can be broken down into subfamilies to help identify where failures occur. While the present work focused on the analysis of existing measures of generalization, future work could build on the robust regression methodology of Appendix B and attempt to formulate new robust measures via gradient-based optimization.

The development of benchmarks and quantitative metrics has been a boon to machine learning. We believe that methodology based on robustness with carefully crafted interventions will best serve our scientific goals.

## Broader Impact

Our work aims to sharpen our understanding of generalization by improving the way that we evaluate theories of generalization empirically. The proposed methodology is expected to aid in the quest to understand generalization in deep neural networks. Ultimately, this could lead to more accurate and reliable models and strengthen the impact of machine learning in critical applications where accuracy must be predictable. We believe that this work has no direct ethical implications. However, as with all advances to machine learning, long-term societal impacts depend heavily on how machine learning is used.

## Funding Sources

LW was supported, in part, by an NSERC Discovery Grant. DMR was supported, in part, by an NSERC Discovery Grant, Ontario Early Researcher Award, and a stipend provided by the Charles Simonyi Endowment. This research was carried out while GKD and DMR participated in the Special Year on Optimization, Statistics, and Theoretical Machine Learning at the Institute for Advanced Study.

## Acknowledgements

The authors would like to thank Grace Abuhamad, Alexandre Lacoste, Gaël Letarte, Ben London, Jeffrey Negrea, and Jean-Philippe Reid for feedback on drafts.

## Footnotes

*Equal contribution. Correspondence to: {karolina.dziugaite, alexandre.drouin}@elementai.com

[2]Jiang et al. also propose a conditional-independence test, which we discuss in Appendix E.

[3]See Appendix E for a comparison to Jiang et al. [9]'s conditional-independence-based approach.

[4]In order to match [9], in all of our experiments, we train to a fixed level of cross entropy loss that also yields zero training error. Since $G = R - \hat{R}$, and $\hat{R} = 0$, a prediction task that depends on changes in generalization error $G$ is equivalent to one that depends on changes in risk $R$.

[5]We apply a $\frac{1 - \Psi}{2}$ transformation to obtain values in $[0, 1]$, where 1 is achieved if $\Psi = -1$ and 0 if $\Psi = 1$.

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
