[Supplementary Material]

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

# Appendix

## Table of Contents

## Contents

## A  Importance sampling schemes to account for Monte Carlo noise

Generalization error $G(\omega)$ is estimated from a held-out test set, since we do not have access to the data distribution. The size of the test set determines the precision at which true generalization error can be approximated. Let $G, G'$ denote true generalization errors for $\omega, \omega'$, respectively, and similarly $\hat{G}, \hat{G}'$ estimates of generalization made on a test set of size $m$. Let $\epsilon = |\hat{G} - \hat{G}'|/2$. Then

$$\mathbb{P}(\mathrm{sgn}(G - G') = \mathrm{sgn}(\hat{G} - \hat{G}')) \geq \mathbb{P}(|G - \hat{G}| \leq \epsilon) \cap (|\hat{G}' - G'| \leq \epsilon)) \tag{4}$$

$$\geq (1 - \mathbb{P}(|G - \hat{G}| \geq \epsilon))(1 - \mathbb{P}(|\hat{G}' - G'| \geq \epsilon)). \tag{5}$$

We can now apply Hoeffding inequality to bound $\mathbb{P}(|G - \hat{G}| \geq \epsilon) \leq 2e^{-2m\epsilon^2}$. Let $\chi(\epsilon, m) = (1 - 2e^{-2m\epsilon^2})^2$. If $\hat{G}$ and $\hat{G}'$ are computed using $m$ samples, then accepting samples only when $\chi(\epsilon, m) > p$ would mean that with probability at least $p$, $\mathrm{sgn}(G - G') = \mathrm{sgn}(\hat{G} - \hat{G}')$.

In our experiments, we only keep the samples with $\chi((\hat{G}(\omega') - \hat{G}(\omega))/2, m) > 0.5$. The expectation in the sign-error in Eq. (3) is approximated by a weighted average of the loss for each sample, where the weight is equal to a rescaled version of

$$\kappa(\omega, \omega') = \max(0, \chi((\hat{G}(\omega') - \hat{G}(\omega))/2, m) - 0.5). \tag{6}$$

## A.1 Filtering environments

The weighting scheme proposed in Eq. (6) allows to downweigh (and in some cases discard) pairs of experiments for which the generalization errors do not differ significantly. Consequently, some environments may be left with very few samples. Let the $i^{\text{th}}$ sample have a weight $\kappa_i$. To avoid calculating the expected sign-error on too few data points, we discard environments where the *effective sample size*, defined as

$$n_{\text{eff}} = \frac{(\sum_i \kappa_i)^2}{\sum_i \kappa_i^2}, \tag{7}$$

is smaller than 12. In our case, the weights are as defined in Eq. (6). The choice of 12 samples means we estimate the expected loss to a precision of around standard deviation divided by 3 ([14][Ch. 29, p. 380]).

In Fig. 3, we show the number of environments remaining at various $n_{\text{eff}}$ cutoffs. Notice that very few environments are included for the width hyperparameter, even at $n_{\text{eff}} \geq 12$. This is because the variations in generalization error due to width are often negligible in our data. Therefore, many of the environments where the width is varied are automatically discarded.

Figure 3: Number of environments remaining (per hyperparameter) at various effective sample size thresholds. The red vertical bar marks $n_{\text{eff}} = 12$, which is the minimum effective sample size that we consider. Notice how most environments where width is varied have small effective sample sizes.

## A.2 Calculating the sign-error in empirical evaluation

Given the weighing and filtering schemes described above, we present the exact formulation of the sign-error used in the coupled-network experiments of Section 6. For each environment $e$, we sample seeds $\{\sigma_i, \sigma'_i\}_{i \in \{1,..,10\}}$, yielding a total of 100 samples $\{\omega_i, \omega'_j\}_{(i,j) \in \{1,..,10\}^{\otimes 2}}$. For the sake of computation, we then use $\mathcal{E} = \{(\omega_i, \omega'_i)\}_{(i,j) \in \{1,..,10\}^{\otimes 2}}$ as our samples from $P^e$. We calculate the empirical sign-error as:

$$
\begin{aligned}
\widehat{\text{SE}}(\mathcal{E}, C) = \frac{1}{2} \mathbb{1}\Big(n_{\text{eff}} \geq 12\Big) \\
\sum_{(\omega, \omega') \in \mathcal{E}} \bar{\kappa}(\omega, \omega') \big[1 - \text{sgn}\left(\hat{G}(\omega') - \hat{G}(\omega)\right) \cdot \text{sgn}\left(C(\omega') - C(\omega)\right)\big],
\end{aligned} \tag{8}
$$

where the effective sample size $n_{\text{eff}}$ is defined in Eq. (7), and $\bar{\kappa}(\omega, \omega') = \dfrac{\kappa(\omega, \omega')}{\sum_{(\omega_i, \omega'_i) \in \mathcal{E}} \kappa(\omega_i, \omega'_i)}$ are

normalized weights.

## A.3 Ablation study: What is the effect of Monte Carlo noise?

In this section, we investigate how filtering Monte Carlo noise using the aforementioned procedure affects the estimated sign-errors. This is done by running an ablation study where the sign-errors

(a) Mean

(b) 90$^{\text{th}}$ percentile

(c) Maximum

Figure 4: Effect of Monte Carlo noise filtering on various statistics of the sign-error distribution for each measure.

are computed with and without Monte Carlo noise filtering and the resulting error distributions are compared. Specifically, for the *with filtering* case, pairs $(\omega, \omega')$ in each environment are weighted using $\kappa(\omega, \omega')$ as described at Eq. (6). Conversely, for the *without filtering* case, pairs are weighted with $\kappa(\omega, \omega') = 1$, allowing pairs with very small differences in generalization gap to be included in the expectation. Intuitively, Monte Carlo noise should increase the sign-error in environments where it occurs, since it can randomly push $\text{sgn}(\hat{G}(\omega) - \hat{G}(\omega'))$ to be $+1$ or $-1$, making it unpredictable.

The results reported in Fig. 4 support this hypothesis: including noisy ranking pairs generally leads to larger sign-errors. Indeed, for many generalization measures, the mean sign-error decreases with noise filtering (Fig. 4a). Moreover, extreme values such as the maximum do not seem to be affected by noise filtering (Fig. 4c), which was expected since noise is likely not prevalent in each of the considered environments. In light of these results, we conclude that the procedure that we propose to account for Monte Carlo noise (Appendix A) is beneficial and that it should be implemented in studies, such as ours and Jiang et al. [9], that rely on ranking comparisons of generalization gaps.

## B Evaluating robust prediction of the numerical value of generalization error

In single-network experiments, we evaluate the ability of *generalization measures* to predict the exact numerical value of the generalization error.

**Evaluation criterion.** We rely on a robust mean squared error (MSE) objective. For a transformation $f_\theta(\cdot)$ of a generalization measure $C$, the robust MSE is

$$\inf_\theta \sup_{e \in \mathcal{F}} \text{MSE}(P^e, f_\theta(C)) \text{ where } \text{MSE}(P^e, C') = \mathbb{E}_{\omega \sim P^e} (C'(\omega) - G(\omega))^2. \tag{9}$$

Choosing $f_a(x) = ax$ in Eq. (9), we recover *robust risk of the linear oracle transformation of a generalization measure $C$*. Similarly, choosing $f_{(a,b)}(x) = ax + b$, we get *robust risk of the affine oracle transformation of a generalization measure $C$*.

**Environments.** In this setting, each environment $e \in \mathcal{F}$ is defined by a single hyperparameter configuration $H$. The data points in each environment are acquired by training a model with hyperparameters $H$ and varying the random seed. For example, an environment could be composed of multiple experimental records where learning rate is 0.01, the model depth is 2, model width is 10, the dataset is CIFAR-10, and the training set size is 50 000. The hyperparameter values considered are those given at Appendix C and we consider ten random seeds, resulting in 1000 environments with ten data points each.

### B.1 Experiments and results

In our experiments, we fit an *affine oracle* for each generalization measure by minimizing the robust mean squared error. This evaluates the ability of each measure to predict generalization error in each environment, up to a common linear rescaling and an additive constant. Note that we constrain the linear coefficients to be non-negative (i.e., $a \geq 0$), since we expect these measures to upper bound generalization error. We compare the performance of the affine oracles to that of a baseline that ignores the generalization measures and only fits a bias parameter (i.e., $a = 0$). The results are reported in Fig. 5.

Figure 5: Cumulative distribution of robust root mean squared error (RMSE) of affine oracles trained on the 1000 environments defined at Appendix B.1. The maximum and average RMSE over all environments are shown in black and orange, respectively. A measure leading to perfect prediction of generalization error would have a white column and a black line at zero. The RMSE of a baseline oracle that only fits a bias parameter is shown in red.

Observe that many generalization measures achieve lower robust mean squared error than the baseline oracle. This suggests that these measures do carry meaningful information about generalization error that transfers across all environments. In addition, notice that many measures that show good robust performance in this setting often perform well in the coupled-network experiments (see Section 6 and Appendix D). For instance, the PAC-Bayes measures based on `flatness` show signs

of robustness (although not perfect) in both types of experiments. Furthermore, measures based on `path.norm`, which show strong signs of robustness in the SVHN-only coupled-network experiments (see Appendix D.1) are also among the best performers in this setting. Finally, notice how the average mean squared error tends to be very similar across measures, while their robust mean squared errors differ more. This provides additional evidence that averaging can mask failures in robustness and that a worst-case analysis should be preferred.

## B.2  Exploring weaker families of environments

In this section, we consider families of environments $\mathcal{F}$ which are intermediate between the robust regression described above and empirical risk minimization (where the average MSE is minimized). Minimizing the robust risk should be an easier task in these environments, due to increasing levels of averaging which mask robustness failures.

**Varying a single hyperparameter.**   In this setting, each environment $e \in \mathcal{F}$ is composed of runs where a single variable $H_i \in H$ is allowed to vary. All other variables $\mathcal{V} \setminus H_i$ are fixed, though the random seed varies. For example, an environment could be composed of all runs where the model depth is 2, model width is 10, the dataset is CIFAR-10, the training set size is $50\,000$, and the learning rate takes any of the 5 values considered. When considering the hyperparameter values described at Appendix C, we obtain 1350 environments with between 20 to 50 points each (due to 10 repeats per hyperparameter setting). The results for all experiments over this family of environments are reported in Fig. 6.

As expected, the range of robust MSE values attained per measure is lower than in Fig. 5, since the added averaging makes the task easier. As in the previous section, we can see that the mean MSE over environments obscures a clear ordering over measures that is present when using the robust MSE. Also, notice that the ordering induced by the robust MSEs is similar to that in Fig. 5, with some PAC-Bayesian measures and `path.norm` measures performing best.

**Varying all but one hyperparameter.**   In this setting, each environment $e \in \mathcal{F}$ is composed of runs where a single variable $H_i \in H$ is fixed. All other variables $\mathcal{V} \setminus H_i$ and the random seed vary. For example, an environment could be composed of all runs where the model depth is 2 and the other parameters take on every possible value. When considering the hyperparameter values described at Appendix C, we obtain 21 environments with 2000 to 5000 points each. The results for all experiments over this family of environments are reported in Fig. 7.

The gap between the mean and robust RMSEs is much narrower here, caused by averaging the MSE over much more points. Nevertheless, we are still able to see a very similar ordering to that of Fig. 5 and Fig. 6 preserved here.

Figure 6: Cumulative distribution of robust root mean squared error (RMSE) of affine oracles trained on the 1350 environments where a single hyperparameter is varied and the others are fixed. Results are shown over all environments (top row) and separately for the subset of environments where each hyperparameter is varied (other rows). The maximum and average RMSE over all environments in each row are shown in black and orange, respectively. A measure leading to perfect prediction of generalization error would have a white column and a black line at zero. The RMSE of a baseline oracle that only fits a bias parameter is shown in red.

Figure 7: Cumulative distribution of robust root mean squared error (RMSE) of affine oracles trained on the 21 environments where one hyperparameter is kept fixed and all the others are varied. Results are shown over all environments (top row) and separately for the subset of environments where each hyperparameter is kept fixed (other rows). The maximum and average RMSE over all environments in each row are shown in black and orange, respectively. A measure leading to perfect prediction of generalization error would have a white column and a black line at zero. The RMSE of a baseline oracle that only fits a bias parameter is shown in red.

# C Experimental details, code, and data

Our experimental protocol is inspired by that used in the large-scale study of Jiang et al. [9]. We proceed in two steps: 1) data collection, where many models are trained with various hyperparameter configurations, 2) data analysis (e.g. coupled-network experiments; see Section 5). Our code relies on PyTorch [21], NumPy [6], Pandas [15], Scikit-Learn [22] and Matplotlib [7].

**Availability:** Our code is open-source and available online, along with the data used in the experiments: https://github.com/nitarshan/robust-generalization-measures

## C.1 Hyperparameters

Each data point in our analysis is obtained by training a model with a given hyperparameter configuration. Between data points, we vary 5 hyperparameters that alter the model, the learning procedure, and the data distribution. These are:

1. Learning rate: $H_{lr} \in \{10^{-3}, 10^{-2.8}, 10^{-2.5}, 10^{-2.2}, 10^{-2}\}$
2. Model depth: $H_{depth} \in \{2, 3, 4, 5, 6\}$
   *This corresponds to the number of "blocks" used in our model architecture. Each block corresponds to 3 convolutional layers.*
3. Model width: $H_{width} \in \{8 \times 25, 10 \times 25, 12 \times 25, 14 \times 25, 16 \times 25\}$
   *This corresponds to the filter width used in convolutional layers of our model architecture.*
4. Dataset: $H_{dataset} \in \{\text{CIFAR-10}, \text{SVHN}\}$
5. Training set size: $H_{size} \in \{50\,000, 25\,000, 12\,500, 6\,250\}$

## C.2 Models

We use a fully convolutional "Network-in-Network" architecture similar to that described in [13] and used for the study in [9]. A full specification of our model can be found in our codebase.

While the most successful model architectures of today employ residual connections between blocks of convolutional layers, we are unable to make use of those here due to the unclear applicability of many bounds to models using skip-connections.

## C.3 Datasets

We make use of two common vision datasets: CIFAR-10 [10] and SVHN [18]. Both datasets are composed of 32x32 RGB images with 10 classes of natural images, with CIFAR-10's classes corresponding to animals and vehicles, and SVHN's classes corresponding to digits cropped from Street View images. We make use of the full training (50k images) and testing (10k images) splits of CIFAR-10, and randomly sample (without replacement) a subset of the larger training and testing sets of SVHN to match the split sizes of CIFAR-10. We also sample without replacement when generating the smaller training sets of size 25k, 12.5k and, 6.25k, but always make use of the same testing sets of size 10k across all experiments.

We do not make use of data augmentation when passing these images into our models, following the observation in [9] that doing so negatively affects the ability of these models to consistently reach low cross-entropies.

## C.4 Training procedure

We use SGD with a momentum parameter of 0.9 for all experiments. We do not use learning rate decay or weight decay. As in [9] we use a cross-entropy stopping criterion, which we set to 0.01 for all experiments, and calculate over the entire training dataset.

## C.5 Data collection

We run 10 repeats with different random seeds for each of the 1000 possible hyperparameter combinations, providing a total of 10,000 experimental runs. Of these, 300 runs failed to meet the

cross-entropy criterion as well as an additional training accuracy criterion of being greater than 99%. These data points were filtered out before any analysis.

As in [9] we "fuse" the batch-norm layers of our model with their preceding convolutional kernels before calculating the value of the generalization measures.

## C.6 Measures

We look at the following 24 generalization measures for convolutional networks, which are modifications of a subset of those studied in [9]. A key difference is that, while those original measures do not account for dataset size, we correct for this through a normalization of the form $\sqrt{C/m}$ for all measures $C$, where $m$ is the size of the training dataset.

While we provide the mathematical expressions for all our measures here, we direct the reader to [9, Appendix D] for more details. It is worth noting, however, that while many of these expressions are derived from generalization bounds, there is no requirement that generalization measures correspond to bounds. Even among the measures that correspond to bounds, direct comparison is not necessarily meaningful. Some are generalization bounds for the networks learned by SGD. Some, in particular those derived from PAC-Bayes methods, are bounds on stochastic classifiers, e.g., obtained by randomizing the weights of a neural network in some way. Some of these quantities control the generalization error between the risk and empirical risk, while others relate to the difference between surrogate risks, e.g., based on margins.

We calculate the spectral norm of convolutional layers using the exact FFT-based method of [26]. We had initially used the approximation proposed in Yoshida and Miyato [28], but our findings were not noticeably different between the approximation and exact methods.

For reasons of numerical stability, we apply log transformations to some of these measures. **As this is a monotonic transformation, it does not affect the ranking results covered in the main paper.**

### C.6.1 VC Measures

For a convolutional network of $d$ layers, with a $k_i \times k_i$ kernel and $c_i$ filters at depth $i$:

$$C_{params} = \sqrt{\frac{\sum_i^d k_i^2 c_{i-1}(c_i + 1)}{m}} \tag{10}$$

### C.6.2 Output Measures

Let $\gamma$ be the 10th-percentile of margin values over the training dataset.

$$C_{inverse.margin} = \sqrt{\gamma^2 m}^{-1} \tag{11}$$

### C.6.3 Spectral Measures

Let $\mathbf{W}_i$ denote the $i^{\text{th}}$ convolutional layer's weight tensor, and $\mathbf{W}_i^0$ it's initial value. Let $\|\mathbf{W}_i\|_2$ denote it's spectral norm. Let $\|\mathbf{W}_i\|_F = \|\text{vec}(\mathbf{W}_i)\|$ denote it's Frobenius norm.

$$C_{log.spec.init.main} = \log \sqrt{\frac{\prod_{i=1}^d \|\mathbf{W}_i\|_2^2 \sum_{j=1}^d \frac{\|\mathbf{W}_j - \mathbf{W}_j^0\|_F^2}{\|\mathbf{W}_j\|_2^2}}{\gamma^2 m}} \tag{12}$$

$$C_{log.spec.orig.main} = \log \sqrt{\frac{\prod_{i=1}^d \|\mathbf{W}_i\|_2^2 \sum_{j=1}^d \frac{\|\mathbf{W}_j\|_F^2}{\|\mathbf{W}_j\|_2^2}}{\gamma^2 m}} \tag{13}$$

$$C_{log.prod.of.spec.over.margin} = \log \sqrt{\frac{\prod_{i=1}^d \|\mathbf{W}_i\|_2^2}{\gamma^2 m}} \tag{14}$$

$$C_{log.prod.of.spec} = \log \sqrt{\frac{\prod_{i=1}^d \|\mathbf{W}_i\|_2^2}{m}} \tag{15}$$

$$C_{fro.over.spec} = \sqrt{\frac{\sum_{i=1}^d \frac{\|\mathbf{W}_i\|_F^2}{\|\mathbf{W}_i\|_2^2}}{m}} \tag{16}$$

$$C_{log.sum.of.spec.over.margin} = \log \sqrt{\frac{d \left( \frac{\prod_{i=1}^d \|\mathbf{W}_i\|_2^2}{\gamma^2} \right)^{1/d}}{m}} \tag{17}$$

$$C_{log.sum.of.spec} = \log \sqrt{\frac{d \left( \prod_{i=1}^d \|\mathbf{W}_i\|_2^2 \right)^{1/d}}{m}} \tag{18}$$

### C.6.4 Frobenius Measures

$$C_{log.prod.of.fro.over.margin} = \log \sqrt{\frac{\prod_{i=1}^d \|\mathbf{W}_i\|_F^2}{\gamma^2 m}} \tag{19}$$

$$C_{log.prod.of.fro} = \log \sqrt{\frac{\prod_{i=1}^d \|\mathbf{W}_i\|_F^2}{m}} \tag{20}$$

$$C_{log.sum.of.fro.over.margin} = \log \sqrt{\frac{d \left( \frac{1}{\gamma^2} \prod_{i=1}^d \|\mathbf{W}_i\|_F^2 \right)^{1/d}}{m}} \tag{21}$$

$$C_{log.sum.of.fro} = \log \sqrt{\frac{d \left( \prod_{i=1}^d \|\mathbf{W}_i\|_F^2 \right)^{1/d}}{m}} \tag{22}$$

$$C_{fro.dist} = \sqrt{\frac{\sum_{i=1}^d \|\mathbf{W}_i - \mathbf{W}_i^0\|_F^2}{m}} \tag{23}$$

$$C_{dist.spec.init} = \sqrt{\frac{\sum_{i=1}^d \|\mathbf{W}_i - \mathbf{W}_i^0\|_2^2}{m}} \tag{24}$$

$$C_{param.norm} = \sqrt{\frac{\sum_{i=1}^d \|\mathbf{W}_i\|_F^2}{m}} \tag{25}$$

### C.6.5 Path Measures

Define the parameter vector as $\mathbf{w} = \text{vec}(\mathbf{W}_1, \ldots, \mathbf{W}_d)$. Below, $f_{\mathbf{w}^2}(\mathbf{1})[i]$ denotes the $i^{\text{th}}$ logit output of a network using squared weights where the input is a vector of ones.

$$C_{path.norm.over.margin} = \sqrt{\frac{\sum_i f_{\mathbf{w}^2}(\mathbf{1})[i]}{\gamma^2 m}} \tag{26}$$

$$C_{path.norm} = \sqrt{\frac{\sum_i f_{\mathbf{w}^2}(\mathbf{1})[i]}{m}} \tag{27}$$

### C.6.6 Flatness Measures

Let $\omega$ denote the number of weights. Let $\epsilon = 1 \times 10^{-3}$. We use the search procedure for $\sigma$ described in [9], where it is chosen to be the largest number such that $\mathbb{E}_{\mathbf{u} \sim \mathcal{N}(0, \sigma^2 I)} \left[ \hat{L}(f_{\mathbf{w}+\mathbf{u}}) \leq 0.1 \right]$. Similarly, we choose the magnitude-aware $\sigma'$ to be the largest number such that $\mathbb{E}_{\mathbf{u}} \left[ \hat{L}(f_{\mathbf{w}+\mathbf{u}}) \leq 0.1 \right]$, where $u_i \sim \mathcal{N}(0, \sigma'^2 |w_i|^2 + \epsilon^2)$.

$$C_{pacbayes.init} = \sqrt{\frac{\frac{\|\mathbf{w}-\mathbf{w}^0\|_2^2}{4\sigma^2} + \log\left(\frac{m}{\sigma}\right) + 10}{m}} \tag{28}$$

$$C_{pacbayes.orig} = \sqrt{\frac{\frac{\|\mathbf{w}\|_2^2}{4\sigma^2} + \log\left(\frac{m}{\delta}\right) + 10}{m}} \tag{29}$$

$$C_{pacbayes.flatness} = \sqrt{\frac{1}{\sigma^2 m}} \tag{30}$$

$$C_{pacbayes.mag.init} = \sqrt{\frac{\frac{1}{4}\sum_{i=1}^{\omega} \log\left(\frac{\epsilon^2 + (\sigma'^2+1)\|\mathbf{w}-\mathbf{w}^0\|_2^2/\omega}{\epsilon^2 + \sigma'^2|w_i - w_i^0|^2}\right) + \log\left(\frac{m}{\delta}\right) + 10}{m}} \tag{31}$$

$$C_{pacbayes.mag.orig} = \sqrt{\frac{\frac{1}{4}\sum_{i=1}^{\omega} \log\left(\frac{\epsilon^2 + (\sigma'^2+1)\|\mathbf{w}\|_2^2/\omega}{\epsilon^2 + \sigma'^2|w_i - w_i^0|^2}\right) + \log\left(\frac{m}{\delta}\right) + 10}{m}} \tag{32}$$

$$C_{pacbayes.mag.flatness} = \sqrt{\frac{1}{\sigma'^2 m}} \tag{33}$$

# D   Coupled-network: Additional experimental results

## D.1   Restricting the analysis to the SVHN dataset

In Section 6, we observed that some measures failed to be robust across changes in depth for CIFAR-10 environments (e.g., `pacbayes.mag.flatness`). It is therefore reasonable to ask what would happen if we restricted the study to SVHN. Thus, we replicate the experiments reported in Fig. 1 of the main text, but leave out all CIFAR-10 environments. The results are reported in Fig. 8.

Notice how all measures still achieve a robust sign-error of 1.0 overall, but that many measures now have a 90th percentile much closer to zero. This indicates that the error distributions of some measures would have been judged to be robust on some large subfamily of environments. Furthermore, observe how some measures now achieve perfect robustness on 'Depth' environments (e.g., `path.norm`), while none had achieved a sign-error lower than 1.0 in Fig. 1.

Our methodology allows to dig deeper into these results. For instance, we can try to understand where `pacbayes.orig` fails to be robust by looking at the CDF of sign-errors for every pair of values of depth (Fig. 9a) and width (Fig. 9b). We observe that most failures in 'Depth' environments occur when varying depth from $2 \rightarrow 3$. The sources of non-robustness in width are slightly harder to interpret, but we still observe that the measure is significantly more robust for some pairs of width than others.

Digging even deeper, we can look at the distribution of hyperparameter values in the environments where this measure fails to be robust. For 'Depth' environments, we observe that there are 10 environments with a sign-error greater than 0.01 and that these all correspond to very wide networks ($\{14, 16\}$) with small learning rates ($\{0.001, 0.0016\}$) trained on a small dataset (6250 examples). For 'Width' environments, we observe 16 environments with a sign-error greater than 0.01 and that these all correspond to shallow models of depth 2, trained with large learning rates ($\{10^{-2.5}, 10^{-2.2}, 10^{-2}\}$) on datasets of less than $50,000$ examples.

This example clearly illustrates how our proposed methodology allows to zero-in on cases where generalization measures fail to be robust. We expect that studying the shortcomings of measures at such a detailed level will aid in elaborating new, more robust, theories of generalization.

Figure 8: Cumulative distribution of the sign-error across subsets of environments for each generalization measure (in SVHN only). The measures are ordered based on the mean across 'All' environments. A completely *white* bar indicates that the measure is perfectly robust, whereas a *dark blue* bar indicates that it completely fails to be robust.

(a) Interventions on depth

(b) Interventions on width

Figure 9: CDFs of sign-errors separated by environments where depth and width vary between two values for `pacbayes.orig` (on SVHN only). The red Xs indicate that no environments remained after accounting for Monte Carlo noise.

## D.2 Restricting the analysis to the CIFAR-10 dataset

Figure 10: Cumulative distribution of the sign-error across subsets of environments for each generalization measure (in CIFAR-10 only). The measures are ordered based on the mean across 'All' environments. A completely *white* bar indicates that the measure is perfectly robust, whereas a *dark blue* bar indicates that it completely fails to be robust.

### D.3 Exploring a weaker family of environments

We further evaluate the performance of generalization measures in a family of environments that is intermediate between the one considered in Section 5 and by Jiang et al. [9]. As described in Section 5, each environment contains pairs of hyperparameter settings where, within a pair, one hyperparameter is varied between two specific values (e.g., depth 2 $\rightarrow$ 3). However, here, the value of the other hyperparameters is allowed to change between the pairs. For example, assuming only two hyperparameters (width ($w$) and depth ($d$)), the pairs $\{[(d = 2, w = 10), (d = 3, w = 10)], [(d = 2, w = 5), (d = 3, w = 5)]\}$ could belong to the same environment (depth 2 $\rightarrow$ 3), whereas this would not be allowed in Section 5. Hence, one environment in this setting corresponds to the union of multiple environments in the setting described at Section 5. Achieving robustness in this family of environments may be significantly easier due to further averaging, which can mask failures of robustness in some hyperparameter configurations.

As illustrated in Fig. 11, this setting appears to be less challenging than the one described in Section 5. In fact, in Fig. 1, we observe that all measures achieve a robust sign-error (max value) of 1. However, in Fig. 11, we observe that some measures, such as `pacbayes.mag.flatness` and `path.norm.over.margin`, achieve a maximum sign-error much lower than 1. It is thus apparent that the further averaging that occurs in this setting leads to an easier task and prevents some failures in robustness of being exposed.

We therefore confirm that the set of environments considered in the main text is more challenging and that it constitutes a more relevant setting for studying the robustness of generalization measures.

Figure 11: Cumulative distribution of the sign-error across an alternative family of environments where the values of only one pair of variables are fixed and the error is averaged over the rest of variable settings. The number of such environments is shown in parenthesis. The measures are ordered based on their robust sign-error (max) across 'All' environments. A completely *white* bar indicates that the measure is perfectly robust, whereas a *dark blue* bar indicates that it completely fails to be robust.

# E    Methodological comparison to the conditional-independence testing method of Jiang et al. [9]

In addition to their Kendall-$\tau$-based $\Psi$ measure, Jiang et al. [9] propose a measure based on conditional-independence testing (Section 2.2.3 in [9]). This measure attempts to identify the existence of a causal relationship (edge in a postulated causal graph) between a generalization measure and the generalization gap. While the concept of robustness (that our work builds on) and conditional-independence testing can both be tied to the causal inference literature, our methods are fundamentally different. Notably, we do not claim, nor seek, to identify causal relationships (see the discussion in Section 2). Below, we highlight some key differences between our approaches.

It can be tempting to see a similarity in the fact that both methods look at extreme values (min and max). Our method looks at the maximum sign-error over all environments, which is analogous to taking the max over every possible intervention on each hyperparameter. The method of Jiang et al. [9] considers the minimum normalized conditional mutual information ($\hat{\mathcal{I}}$; Eq. (13) of [9]) over all conditioning sets of two hyperparameters (Eq. (14) of [9]). However, as described in their Eq. (11) and (12), the calculation of $\hat{\mathcal{I}}$ involves *averaging over all values* of the hyperparameters in the conditioning set (i.e., $\sum_{U_\mathcal{S}} p(U_\mathcal{S})$). Therefore, while they may appear similar, both approaches are fundamentally different in that one averages over multiple values of the same hyperparameter, while the other (ours) does not.

Furthermore, in the limit where we observe every possible intervention on the HPs, our method identifies measures that may have a causal relationship to generalization (all causal explanations are necessarily robust in this extreme case). However, this is not necessarily true for the IC-based method of Jiang et al. [9]. The reason is that their conditioning sets are of size 2, which may leave open confounding paths in the graph if more than 2 hyperparameters act as confounders, resulting in non-causal mutual information. This means that the method is not guaranteed to detect a causal edge from the generalization measure to generalization gap unless they condition on all hyperparameters. However, if they were to condition on all hyperparameters, their conditional mutual information would collapse to zero, suggesting that there is no causal edge.