[Reviews · NeurIPS 2020]

Review 1

Summary and Contributions: Authors empirically evaluate various generalization bounds via population-of-environments approach, as was done in a recent work [6]. Critical difference from [6] is that the authors use a worst-case focused metric, instead of a correlation-focused metric (e.g. Kendall correlation coefficient in [6]). In particular, they look at what they call the robust sign-error. Empirical evaluation using this metric reveals several interesting findings.

Strengths: - Significance (of the effort): As was the case of [6], the computational effort put into the work is amazing; the experimental results per se may be a great asset to the future researchers, especially if all the trained models, codes, and detailed setups are disclosed to the public. - Significance (of the problem): The problem to be addressed (if is addressed correctly) is definitely significant. Putting aside the importance of understanding the generalization of neural networks models, answering the question "what is the right measure to empirically evaluate theoretical generalization bounds?" is of critical importance.

Weaknesses: - Soundness of the claim (theoretical grounding): Although I have some experience regarding the distributionally robust optimization, I must say that I cannot directly see *why* the proposed "robust error" (which, in my opinion, should rather be called worst-case error), is a better empirical quantity to look at. What properties of the estimate make it so? How should the discrepancy between average-based evaluations and worst-case based evaluations be compared? The authors do refer to some existing works (I am not sure if they are that relevant though), but they should also be formally addressed in the main text as well.

Correctness: Mixed here. The proposed evaluation metric looks reasonable, but I cannot find any concrete reasoning.

Clarity: I do not believe so. - Many claims are ill-cited. For example, see line 47. There, authors state that a "test-set bound provides a sharp estimate of risk," but what exactly do you mean by the "test-set bound"? I guessed that authors are trying to invoke a prediction-theoretic argument, but I could not really be sure because the authors did not make a reference to an example of such test-set bounds. - I also believe that the paper could improve in terms of organization as well. For instance, I needed to spend some time crawling through the paper to find an (explicit) distinction between the environments "e" and the samples "omega" (by the way, are 5 seeds enough?).

Relation to Prior Work: The distinction in terms of the evaluation metric has been made quite clear (although the authors could also introduce the Kendall correlation coefficient formally to help the readers). On the other hand, in terms of discussion of experimental results, I wish to see more explicit comparisons to the results in [6].

Reproducibility: No

Additional Feedback: Response read; participated in the earlier discussion, but forgot to edit the review. Still, I am not fully convinced of the proposed methodology, which is only poetically justified by the authors as 'theory is only as strong as its weakest link.' I strongly believe that understanding and formally explaining why such methodology is better than the previous approach is an essential part of the research procedure. -------------------- - I must recommend renaming some elements, e.g. to have omega \in Omega, f \in \mathcal{F}. P^e looks quite similar to "probability of error" instead of P parametrized by e.


Review 2

Summary and Contributions: - The paper proposes a systematic, empirical methodology to evaluate existing generalization bounds (or measures) with the aim of understanding and more importantly explaining generalization. The main idea constitutes testing the predictions made by a generalization theory (against empirically observed generalization errors) by intervening on relevant characteristics, thus attempting to uncover causal relationships between these characteristics and generalization properties. The importance of such a methodology is highlighted by demonstrating, anecdotally, that generalization bounds can rely on non-causal (but correlated) factors and due to the presence of various such interactions, using these bounds to explain generalization can be complicated. Finally, some findings from the large scale empirical study are presented and briefly discussed. --------------------------------------------- I would like to thank the authors for addressing the questions raised in the review.

Strengths: - The paper is very well written and has good clarity both in terms of readability and conceptual clarity. The motivating example of the SVM's was quite helpful in clarifying the reasoning of the paper. - One central contribution of the paper: using the framework of distributional robustness and considering robust sign error follows from the objective of the paper - evaluating theories of generalization by testing their predictions. This, although it seems like a minor change, crucially differentiates this methodology from other large scale empirical evaluation approaches.

Weaknesses: - A bit more discussion on the findings of the experiments could have been more instructive. -------------------- Upon further discussions and more reflection, I believe that the missing discussion on how to interpret the results is somewhat more concerning than I previously believed. To reflect my updated stance on the paper, I am changing the overall score to a 7.

Correctness: Yes, to the extent that I could verify.

Clarity: Yes, it is one of the main strengths of the paper.

Relation to Prior Work: Yes.

Reproducibility: Yes

Additional Feedback: - Did the authors consider using the approach to formulate robust generalization measures (for the same family of environments used in the paper)? - In the discussion on robustness to width: the authors mention that since the networks are overparameterized, robustness w.r.t width could not be observed. However, shouldn't it be possible to consider a broader range of widths in the experiment to avoid failure due to overparametrization?


Review 3

Summary and Contributions: This paper proposes an improved framework for evaluating theories of generalization for deep neural networks building on the notion of distributional robustness -- in particular, robustness of a theory’s efficacy in response to changes in the hyperparameters. The paper also includes large scale empirical studies that show, under this framework, no existing generalization theory can reliably predict the empirical performance of deep neural networks. Finally, the paper analyzes the advantage and weakness of various families of complexity measures.

Strengths: I enjoyed reading this paper. Like other disciplines of science (e.g. physics), a theory from the first principle is only useful when it fits what we observe in nature, and I believe this “experimental” spirit is missing at large in the theory community of machine learning. I believe the methodology proposed in this paper will help bridge that gap. Distributional robustness over different environments in my opinion is a good way to quantify how good a theory is, and should probably be adopted by all theory, in particular frequentist, bounds papers once suitable benchmarks are established and the details of the applications are further refined. I believe the method is also more flexible than its predecessors and reduces the amount of compute required. This paper is of great importance to the NeurIPS community since generalization is the crux of supervised machine learning.

Weaknesses: I think the claim on the scale made in the paper may be a bit misleading since the actual hyperparameters they search are not that large and do not include common techniques such as dropout or weight decay, but it might be okay since there is no reason why the techniques cannot be applied to any hyperparameters. Another potential issue I see is the relation between this paper and Jiang et al. While the authors claim that Jiang et al. falls short in being distributionally robust, I believe the conditional independence test is extremely close to the method proposed here. In fact, the method has formal connection to intervention and the IC algorithm which is used to build causal graphs. Specifically, Jiang et al also takes a minimum over all possible interventions, which to me seems to be equivalent to taking the infimum over the “environment” proposed in this paper. I hope that the authors can provide a better explanation on how this method improves over the independence test in Jiang et al and outlines the pros/cons of using this method over the former. Some potential benefits I see are that the proposed method is more flexible than the one proposed by Jiang et al due to the introduction of Monte Carlo noise and the ability to define distribution of environment is not only more flexible but also facilitates analysis. That being said, I hope to hear the authors’ thoughts on this. Lastly, it’s not clear to me how the single-network experiment relates to the distribution robustness.

Correctness: I believe that the claims and methods made in the paper are correct.

Clarity: The clarity of the paper could be improved. The definition of environments in section 5 is somewhat hard to follow, and it’s not immediately clear what Monte Carlo noise is referring to. The introduction of measure in the second paragraph of 4.2 and the rest of that paragraph are also not very clear and it’s not clear why it is important. I believe that this paper is not only important to the theory community but also practitioners, so many notations in the paper need to be better explained.

Relation to Prior Work: The discussion on Jiang et al. could be elaborated. See weakness for details.

Reproducibility: Yes

Additional Feedback: ------------------------------------Update----------------------------------- The reviewers and AC had extensive discussion on this paper. I maintain my evaluation so I did not update the score but I believe it's nonetheless good for the authors to see this. I believe the authors have some misunderstanding about (6) and (7) of Jiang et al. which is actually not computing the average case. Regarding MI collapsing to 0, I believe it's not "bug" but a feature. If I know every single thing about the neural networks except for the training randomness, then I hope everyone agrees that repeated training the model would yield models with more or less the same performance, which implies that the MI would be 0. In the light of this, it's actually possible to see the method proposed in this paper as a stronger version of the IC algorithm presented but a weaker version of IC algorithm with all conditional variable. My impression of the paper is positive but I hope the authors can properly address this in the future.


Review 4

Summary and Contributions: The paper proposes a new way to evaluate generalization measures that targets behavior in the worst case rather than on average. The method is applied to study how different generalization measures can prediction impact of hyperparameters. The main experimental setup called coupled-network uses a pair of networks trained independently with all hyperparameters shared except for one. The claim is that a good generalization measure will move the same direction as the generalization error for all perturbations. They conduct a large scale evaluation of a diverse set of measures on CIFAR and SVHN datasets.

Strengths: The paper provides a few surprising findings and insights: - No measure is robust, i.e., they disagree with error for some permutation - num.params in the best measure on average - digging into failure cases for this measure allows spotting failures of the generalization measures that are not visible for non-robust evaluation We find that no existing complexity measure has better robust sign-error than a coin flip. Even 114 though some measures perform well on average, every single measure suffers from 100% failure in 115 predicting the sign change in generalization error under some intervention. This observation is not 116 the end of the evaluation, but the beginning. 117 To better understand the measures, we evaluate them in families of environments defined by inter- 118 ventions to a single hyperparameter. We find: (i) most, though not all, measures are good at robustly 119 predicting changes due to training set size; (ii) robustly predicting changes due to width and depth 120 is hard for all measures, though some PAC Bayes-based measures show (weak) signs of robustness. 121 (iii) norm-based measures outperform other measures at learning rate interventions.

Weaknesses: While the theoretical motivation behind using $sup$ is clear, in practice it makes the evaluation criterion less robust to the coarseness of the underlying set of environments. E.g., if there were only 2 values for width, the results would probably behave quite different. The authors mention that the reason for network width to have little effect is that the neural network is overparametrized. So it's natural to wonder how would the measures perform on bigger datasets such as ImageNet.

Correctness: The methods are clear, but better argumentation for the selection of hyperparameters is preferred (see above).

Clarity: The paper is written well.

Relation to Prior Work: The contribution is clear

Reproducibility: Yes

Additional Feedback:


Review 5

Summary and Contributions: [ QUESTION-ONLY NON-REVIEW FROM AC. SCORE IS FAKE. BUT PLEASE ANSWER! ] 1. I briefly checked your anonymous code and I believe you are calculating spectral norms incorrectly: the spectral norm of a convolution layer is _not_ the spectral norm of its parameters (which define the filter). It is okay to use an approximation, but it should be spelled out without reading the code; meanwhile, to my taste, the spectral norm of the filter is not a reasonable approximation, unless you provide some evidence. Can you please clarify? 2. I realize you are relying on [6] as a source of your generalization measures, but can you at least explain a little about the bounds, for the sake of the reader? Many of these bounds are apples-to-oranges and some discretion is needed to interpret a total ordering of them. E.g., they use varying amounts of information, computation, and bound different things (some are on networks, some are on posterior averages over networks, etc.).

Strengths: .

Weaknesses: .

Correctness: .

Clarity: .

Relation to Prior Work: .

Reproducibility: Yes

Additional Feedback: .

[Author Response · NeurIPS 2020]

We thank the reviewers for their thoughtful feedback. We were pleased to see that several reviewers agree with our proposal to use distributional robustness to evaluate theories of generalization. Several reviewers also acknowledge the "significant computational effort" behind our empirical study and the "surprising findings" we uncover by going beyond average case.

**R1 I cannot [...] see why [...] robust error is a better empirical quantity to look at.** We argue that a theory that only works well on average does not explain generalization. As was presented in Sec. 1 and eloquently summarized by **R3**, a theory must "fit what we observe in nature". Because a theory is only as strong as its weakest link, we propose to use the framework of distributional robustness. In Sec. 6, we show that a focus on worst-case performance allows us to identify failure modes of existing generalization measures that would not have been revealed by an average-case analysis. Each of **R2**, **R3**, **R4** touch on the important difference between average and worst-case analysis here.

**R3 Relationship to the Inductive Causation approach in Jiang et al. (2020)** It is important to clarify key differences between the two approaches. Jiang et al. calculate the normalized conditional mutual information (Eq. (8) in [6]) and take the min over all conditioning sets of two hyperparameters (Eq. (9) in [6]). This is not equivalent to a min over all possible interventions: interventions correspond to setting the variables in the conditioning set to a specific value and their Eq. (6) and (7) take an *average over all values*. This is in sharp contrast with our method, which does measure robustness over all possible interventions. Furthermore, in the limit where we observe every possible intervention on the HPs, our method admits a causal explanation (see Sec 1.1). However, this is not necessarily true for the IC-based method of Jiang et al. The reason is that their conditioning sets are of size 2, which may leave open paths in the graph if more than 2 HPs act as confounders, resulting in non-causal mutual information. This issue cannot be avoided, since their measure collapses to 0 when all HPs are conditioned upon. Finally, our method also allows us to account for Monte Carlo noise in pairwise comparisons and avoids making graphical assumptions about variable relationships.

**R1 R2 R3 R4 Choice of hyperparameter ranges and over/under-parametrization** We agree that studying other datasets, more HPs, underparametrized networks, etc. is interesting. Yet, we feel the scope of our experiments provides sufficient evidence for the merits of our philosophical arguments and methodology in support of robust analysis.

**R4 While the theoretical motivation behind using $\sup$ is clear, in practice it makes the evaluation criterion less robust to the coarseness of the underlying set of environments (e.g., 2 values for width $\rightarrow$ results different).** Average-case analysis seems "more robust" in this way, but this is a deceptively benign property. If the boundaries of a robust analysis change, the conclusions may change because the *implied scope of the theory has changed*. In our analysis and supplement, we argue that one should dig into failures to localize them (see, e.g., Fig. 2 and App. D).

**AC Spectral norm approximation method** In short, thank you: we now use the same method as Jiang et al. (2020). In detail: Approximating the layer spectral norm $||\text{Conv}||_2$ with the reshaped filter spectral norm $||W'||_2$ (where $W'$ is of shape $(c_{out} \times (c_{in} \times h \times w))$) was proposed by Yoshida and Miyato (2017; Sec. 3.2) and used for empirically stabilizing GAN training in Miyato et al. (2018). This method was also used in Neyshabur et al. (2017) and was favourably benchmarked in Sedghi et al. 2018. That said, estimates (such as by Tsuzuku et al. (2018; Cor. 1)) are too loose for comfort and so we now follow Jiang et al. (2020), who use the method of Sedghi et al. (2018), which exactly calculates the spectral norm of a convolutional layer. We have modified our code to use this method instead, and observe that the approximation was fairly accurate and our analysis does not change due to this modification. We will provide a comparison between the approximation and the exact method in the updated supplementary. Hurrah for peer review.

**R3 It's not clear to me how the single-network experiment relates to the distribution robustness.** Generalization measures are expected to predict the exact numerical value of generalization error up to a constant rescaling factor and additive term. We apply robust regression (see Ben-Tal et al. (2009)) to learn constants that hold in all environments.

**AC Give intuition for measures** We agree that this is important and will update the paper accordingly.

**R1 R2 R3 Extended discussion of findings and comparison to Jiang et al.** We want to reiterate that the average-case results that we report are, up to the particular HP value choices, equivalent to the $\Psi$ results reported in Jiang et al. [6]. We can expand on how our reproduction of their average-case study agrees with theirs in the additional page available for the camera-ready.

**R2 Did the authors consider using the approach to formulate robust generalization measures?** Absolutely, this is an interesting future work that we will mention in the camera-ready version.

**R1 Reproducibility: public release of all the trained models, codes, and detailed setups** We believe there is a misunderstanding: all experimental details—code, hyperparameter ranges, and data—were provided at submission (see App. C). We also plan to make a public release of the data and trained models upon acceptance.

**R1 What exactly do you mean by the "test-set bound"?** Apologies. We simply meant an estimate of risk using held-out data (e.g., a test set), using a simple average. We will address this and other clarity issues you raised.

[Meta-Review · NeurIPS 2020]

This paper evaluates various "generalization measures" --- numbers computed from training data and training algorithm and network properties --- in terms of their success predicting generalization. The work builds on the prior work of Jiang et al. (their [6]) in ways they clearly define and thus they provide a new set of results on similar questions. Their changes are interesting, and since generalization of deep networks is of such extensive interest to so many, I also feel these results will be valuable. I look forward to seeing this paper appear, and support the authors on future work. --- Minor comments (just my own opinions and I tried not to use them too much in evaluation). (a) Thanks for trying the spectral thing, it's pretty random I noticed; I've also done many such experiments and was surprised it didn't matter so much in my case, but your metrics are more sensitive than what I've tried, so tbh i think it's quite interesting, maybe worth a mention in an appendix? (b) I still feel (this was my other point) that it would be valuable to especially outside readers to include some table and/or description of these generalization measures, what they mean both rigorously and intuitively, etc. (c) I personally still find the presentation of figure 1 quite dense. Since IMO figure 1 is the main core of this paper, I think it would be reasonable to spend more time explaining figure 1 and even expanding it, in the process shortening some other stuff and moving to appendices? (d) in your feedback, you included responses to reviewers about how your metrics compare to [6]. I think it is essential to include these comments and more in your revisions. (e) your rebuttal was pretty thorough, thanks; I should have highlighted the importance of (b) more, it mattered more to me than (a).